# Regional Attention for Shadow Removal

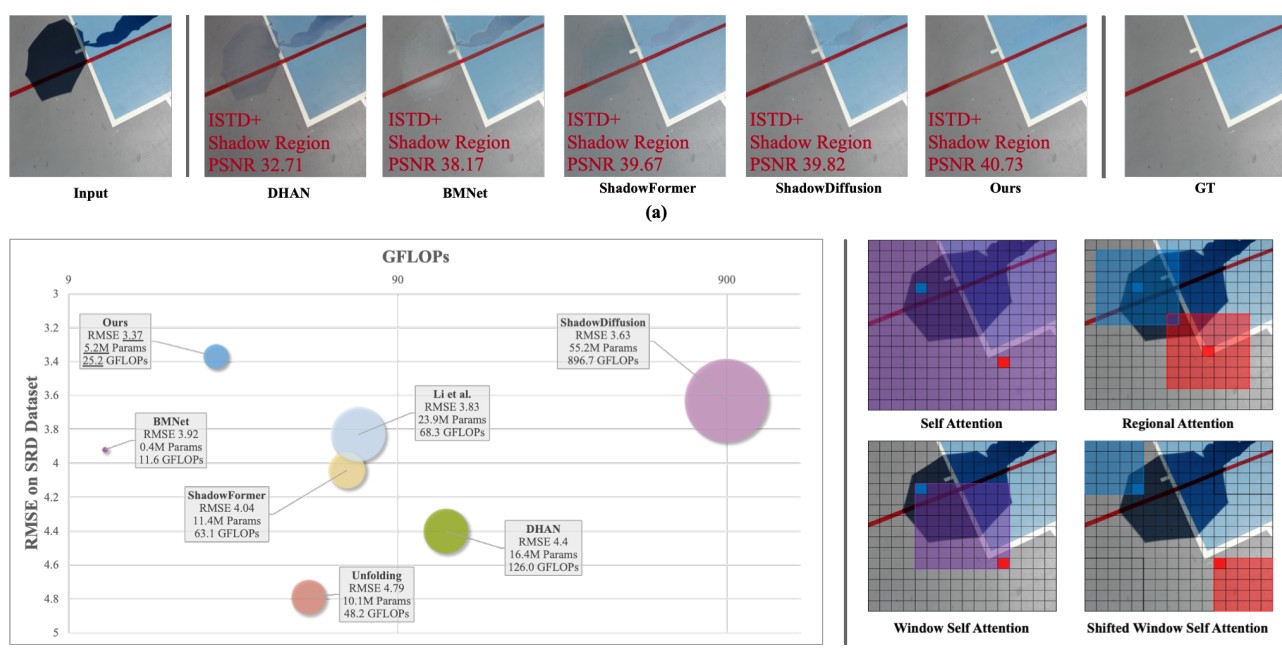

**Figure 1: (a) Performance comparison with previous SOTA methods. Our method achieved a 40.73dB PSNR on the shadow region of the ISTD+ dataset, surpassing the previous SOTA method by 0.90dB; (b) Efficiency comparison with previous SOTA methods. Our method is fast and lightweight with SOTA performance on the SRD dataset; (c) Illustration of self-attentions in shadow removal. Self-attention (used in [12, 20]) has global information exchangeability but with high computational costs. To reduce the complexity, (shifted-)window attention (in [11, 35]) only exchanges the information within a pre-defined cell, but may miss useful clues. Our regional attention refines each token with its neighborhoods, reaching a good balance between effectiveness and efficiency.**

## ABSTRACT

Shadow, as a natural consequence of light interacting with objects, plays a crucial role in shaping the aesthetics of an image, which however also impairs the content visibility and overall visual quality. Recent shadow removal approaches employ the mechanism of attention, due to its effectiveness, as a key component. However, they often suffer from two issues including large model size and high computational complexity for practical use. To address these shortcomings, this work devises a lightweight yet accurate shadow removal framework. First, we analyze the characteristics of the shadow removal task to seek the key information required for reconstructing shadow regions and designing a novel regional attention mechanism to effectively capture such information. Then, we customize a Regional Attention Shadow Removal Model (RASM, in short), which leverages non-shadow areas to assist in restoring shadow ones. Unlike existing attention-based models, our regional attention strategy allows each shadow region to interact more rationally with its surrounding non-shadow areas, for seeking the regional contextual correlation between shadow and non-shadow areas. Extensive experiments are conducted to demonstrate that our proposed method delivers superior performance over other state-of-the-art models in terms of accuracy and efficiency, making it appealing for practical applications. Our code will be made publicly available.

## CCS CONCEPTS

• **Computing methodologies** → **Image processing**; **Computational photography**.

## KEYWORDS

Shadow removal, Regional attention

*MM '24, October 10–14, 2022, Melbourne, Australia*
© 2024 ACM.
ACM ISBN 978-1-4503-XXXX-X/18/06
https://doi.org/XXXXXXX.XXXXXXX

**ACM Reference Format:**
Anonymous Author(s). 2024. Regional Attention for Shadow Removal. In *Proceedings of the 32nd ACM International Conference on Multimedia (MM '23), October 28 - November 1, 2024, Melbourne, Australia.* ACM, New York, NY, USA, 9 pages. https://doi.org/XXXXXXX.XXXXXXX

## 1 INTRODUCTION

When light interacts with objects, shadows are cast. In some cases, shadows can enhance the photography aesthetics. While in others, they act as an interference factor to image quality [38], which may degrade the performance of various vision and multimedia algorithms [30, 39], such as object detection and recognition, and image segmentation. Although this problem has been drawing much attention from the community with significant progress over last years, it still remains challenging for practical use. Because the shadow removal often serves as a preprocessing step for downstream tasks and, more and more systems prefer to deal with images on portable devices anytime and anywhere, besides the high accuracy, its computational cost and model size are expected to be marginal, especially when the computation and memory resources are limited. In other words, a satisfactory shadow removal model shall take into consideration all the removal quality, the processing cost and the model size simultaneously.

In the literature, a variety of shadow removal methods [6–8, 11, 13, 19] have been proposed over last years, aiming to mitigate the negative impact of shadows on image quality and enhance the performance of vision algorithms. Traditional approaches heavily rely on hand-crafted priors, *e.g.*, intrinsic image properties, which are often violated in complex real-world scenarios, and thus produce unsatisfactory results. Deep learning techniques [1, 3, 9, 12, 17, 18, 20] have emerged as powerful alternatives, enabling more robust and data-driven approaches to shadow removal. However, most existing advanced shadow removal models barely consider severe model stacking, necessitating substantial computational resources. This issue significantly limits their applicability to potential downstream tasks in real-world scenarios.

Let us take a closer look at the target problem. Given an image, shadows typically occupy a part of the image. The goal is to convert involved shadow regions into their non-shadow versions, which should be visually consistent with the non-shadow surroundings in the given image. A natural question arises: is all the information in the entire image equally important for the reconstruction of regions affected by the shadow? Intuitively, aside from the darker color of shadowed areas, the most direct way to discern shadows is by the contrast between the shadowed regions and their neighbor non-shadow areas. From this perspective, we can reasonably assume that the critical information for repairing a certain shadow region should be largely from non-shadow areas around the aim region. Based on this assumption, we propose a novel attention mechanism called regional attention, and customize a lightweight shadow removal network, *i.e.*, RASM. Our design is capable of balancing the efficiency and the accuracy of shadow removal in an end-to-end way. As schematically illustrated in Fig. 2, the proposed RASM distincts from previous global and local attention methods, which can reduce computational burden and enhance the rationality of information aggregation between restricted non-shadow and target shadow areas. Experimental comparisons on widely-used shadow

removal datasets (ISTD+ [24] and SRD [32]) and ablation studies reveal the efficacy and superior performance of RASM over other SOTA methods in terms of the effectiveness and the efficiency.

The primary contributions of this paper are summarized as:

- We rethink the key to shadow removal and propose that the information from the regions surrounding the shadows is essential for effective shadow removal. Inspired by this insight, we introduce a regional attention mechanism that allows each shadowed area to aggregate information from its adjacent non-shadowed regions.
- We develop a shadow removal network, RASM, based on a novel regional attention mechanism that optimizes the interaction between shadowed and non-shadowed areas, effectively balancing accuracy and computational efficiency.
- The comprehensive experimental evaluations conducted on the widely recognized ISTD+ [24] and SRD [32] datasets demonstrate that our proposed method achieved a new state-of-the-art performance with a lightweight network architecture.

## 2 RELATED WORK

Traditional methods for shadow removal rely on image properties such as chromaticity invariance [7, 8], gradient consistency [6, 13] or human interactions [10]. The early work in this category can be traced back to [7, 8], where illumination-invariant images are extracted using a pre-calibrated camera. However, the calibration process is laborious, which limits its practical application. To address this issue, work [6] proposes extracting illumination-invariant images through an optimization procedure that does not require any provenance information of the image. Work [13] aims to develop more sophisticated models for the lit and shadowed areas; however, their methods sometimes fail due to the complexity of shadow generalization and imaging procedures. Some works [13, 22] attempt to address this issue by dividing the shadow removal task into two subtasks: shadow detection and mask-based shadow removal. However, since these shadow detectors rely solely on hand-crafted priors, the removal modules may be affected by inaccurate detection results, which can negatively impact overall performance. Furthermore, traditional methods for removing shadows still encounter difficulties when dealing with complex distortions in real-world scenarios, particularly in the penumbra area.

Deep learning has enabled significant advancements in data-driven methods, with works [4, 19] investigating unpaired shadow removal training using generative models. However, these models are typically heavy as they model the distribution of shadow-ed images in a generative manner. In parallel, using a large-scale dataset, Qu et al. [32] were among the first to train an end-to-end deep network for recovering shadow regions. Wang et al. [33] proposed the ISTD shadow removal dataset, featuring manually-annotated shadow masks. However, the non-shadow area of shadow images exhibits substantial inconsistency with the corresponding shadow-free images, as noted in [24], which proposed mitigating this by transforming the non-shadow area of ground-truths using linear regression. Subsequent studies [1, 3, 9, 17, 18, 29] have explored shadow removal with given shadow masks. Recently, some works have sought to improve the computational efficiency of shadow

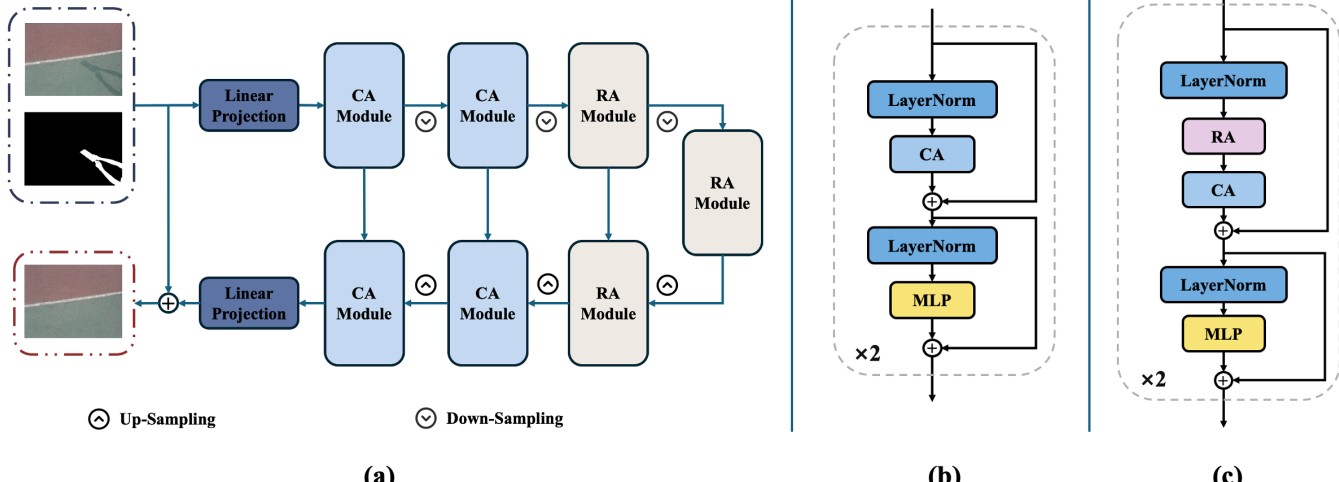

(a)  (b)  (c)

Up-Sampling  Down-Sampling

Figure 2: An illustration of our proposed framework. (a) Overview of the RASM structure. (b) Channel Attention Module. (c) Regional Attention Module. RASM employs the Channel Attention Module for global information interaction, followed by a Regional Attention Module for spatial information interaction.

removal. Zhu et al. [41] proposed a deep unfolding framework for removing shadows efficiently. With the great help of normalizing flow, which reuses the encoder as a decoder, work [40] significantly reduced parameter size. However, representation power was also limited since only half the features were used per layer to ensure invertibility. To address the issue of unsatisfactory boundary artifacts persisting after restoration, Guo et al. [12] proposed the first diffusion-based shadow removal model, which can gradually optimize the shadow mask while restoring the image. Work [21] leverages features extracted from Vision Transformers pre-trained models, which unveil a removal method based on adaptive attention and ViT similarity loss. However, such diffusion-based methods incur significant time and space complexity, leading to significant computational overhead. ShadowFormer [11] proposed re-weighting the attention map in the transformer using the shadow mask to exploit global correlations between shadow and non-shadow areas. However, unlike ShadowFormer, our emphasis lies on the information from non-shadow regions closely surrounding the shadows. We enable each shadow region to integrate information from its immediate non-shadow surroundings. This strategy offers greater flexibility than the window attention used by ShadowFormer and is more aligned with the intrinsic characteristics of shadow removal tasks. Our method achieves superior results without increasing the complexity inherent in ShadowFormer.

## 3 METHODOLOGY

### 3.1 Problem Analysis

In everyday situations, determining the location of shadowed regions often relies on the contrast between the shadowed and adjacent non-shadowed areas. Shadows affect specific areas within an image, resulting in significant differences in brightness, color, and texture compared to non-shadowed surroundings. These differences contain the crucial information required for shadow removal. Sharp transitions in lighting at the periphery of shadowed regions

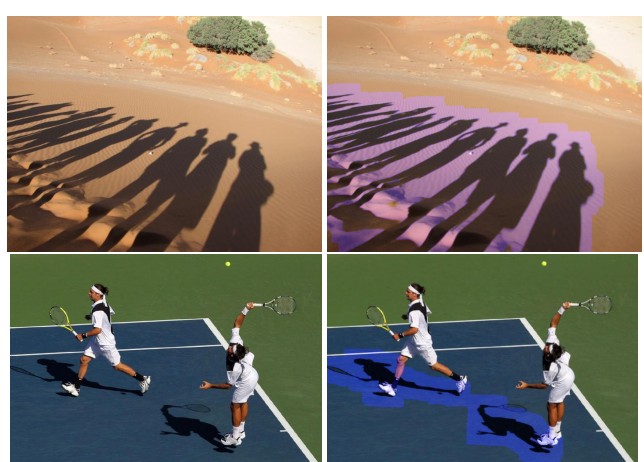

Figure 3: The first column of images presents scenes with shadows. The highlighted regions in the second column of images represent the non-shadowed areas immediately adjacent to the shadows. We posit that the information from these areas is crucial for the task of shadow removal.

usually create distinct gradation zones. Therefore, it is crucial to comprehend and utilize the information from the non-shadowed regions surrounding the shadows to achieve accurate shadow removal, shown in Fig. 3. The importance of the information from non-shadowed regions increases with proximity to shadowed areas. Analyzing the characteristics of these areas not only allows for the accurate identification of shadow boundaries but also provides the necessary reference data for subsequent shadow restoration. Regional attention mechanisms enable models to focus on the critical areas surrounding the shadows, distinguishing which features are important for the task at hand, thereby facilitating more effective information integration.

## 3.2 Regional Attention for Shadow Removal

Contrary to prior studies such as those by [26] and Wang et al. [35], which predominantly address image restoration tasks characterized by global corruption, shadow removal presents a distinct challenge due to its nature of partial corruption. In this context, the non-shadow regions are of critical importance, as they play an essential role in the restoration of shadow-affected areas.

Shadow removal tasks necessitate a large receptive field to assimilate surrounding contextual information effectively. This requirement is rooted in the primary method of distinguishing shadow regions by contrasting them with adjacent non-shadowed areas. Therefore, the primary goal during restoration is to ensure that the reconstructed shadow regions closely resemble their surrounding non-shadow areas, rather than relying excessively on long-range global spatial information. This is where regional attention becomes pivotal, focusing on the information from neighboring regions to enhance the restoration process. The window attention mechanism [26], which projects queries, keys, and values from the information within a specific window to perform self-attention, is less effective for shadow removal. This limitation arises because the non-shadow information required to address shadow discrepancies varies significantly across different locations. Unlike the Window Attention mechanism utilized in prior works [11, 26], which generalizes the attention within a window, our proposed regional attention mechanism is specifically designed to tailor the attention to the specific needs of each shadowed area. By doing so, it ensures that each shadow region can access and integrate distinct and locally relevant non-shadow information, thus facilitating more precise and context-aware restoration.

To fulfill the outlined objectives, we have devised a Transformer-based network leveraging regional attention, termed RASM, specifically for shadow removal. Predicated on our hypothesis that non-shadow information nearer to shadows is of heightened importance, we allocate a minimal proportion of parameters and computational resources to rapidly process global feature interactions, thus allowing greater computational resources and parameters to be concentrated on regional attention processes. RASM operates as a multi-scale encoder-decoder model. Initially, we utilize Channel Attention Module(CA Module) [16] to effectively capture global information. Subsequently, we introduce a module equipped with regional attention, which harnesses spatial and channel-wise contextual information from non-shadow areas to facilitate the restoration of shadow regions during the bottleneck phase.

3.2.1 *Overall Architecture.* For a shadow input $I_s \in \mathbb{R}^{3 \times H \times W}$ accompanied by a shadow mask $I_m \in \mathbb{R}^{H \times W}$, a linear projection LinearProj($\cdot$) is initially applied to generate the low-level feature embedding $X_0 \in \mathbb{R}^{C \times H \times W}$, with $C$ representing the embedding dimension. Subsequently, $X_0$ is processed through an encoder-decoder framework, each composed of CA modules designed to integrate multi-scale global features. Within each CA module are two CA blocks and a scaling layer—specifically, a down-sampling layer in the encoder and an up-sampling layer in the decoder, as depicted in Fig. 2. The CA block functions by compressing spatial information through CA and subsequently capturing long-range

correlations using a feed-forward MLP [5], structured as follows:

$$\tilde{X} = \text{CA}(\text{LN}(X)) + X, \tag{1}$$

$$\hat{X} = \text{GELU}(\text{MLP}(\text{LN}(\tilde{X}))) + \tilde{X}, \tag{2}$$

where LN($\cdot$) denotes the layer normalization, GELU($\cdot$) denotes the GELU activation layer, and MLP($\cdot$) denotes multi-layer perceptron. After passing through $L$ modules within the encoder, we receive the hierarchical features $\{X_1, X_2, \ldots, X_L\}$, where $X_L \in \mathbb{R}^{2^L C \times \frac{H}{2^L} \times \frac{W}{2^L}}$. We calculate the regional contextual correlation via the Regional Attention Module (RAM) according to the pooled feature $X_L$ in the bottleneck stage. Next, the features input to each CA module of the decoder is the concatenation of the up-sampled features and the corresponding features from the encoder through skip-connection.

3.2.2 *Regional Attention Module.* Given that shadow removal is a task of partial corruption, existing local attention mechanisms [26, 35] face considerable constraints during the shadow removal process, as the areas within a window may be entirely corrupted. While [11] mitigates this to a certain extent, it is still limited by the constraints of window-based attention, lacking the flexibility to provide each shadow region with unique, spatially relevant non-shadow information. To address this, we propose a novel Regional Attention Module (RAM), which enables each shadowed location to more effectively utilize regional attention information across spatial and channel dimensions.

In the development of our Regional Attention Module, we draw inspiration from the Neighborhood Attention Transformer [15], adapting its core concepts to better suit the specific challenges of shadow removal. Given a feature map $Y \in \mathbb{R}^{C \times H' \times W'}$ normalized by a LayerNorm (LN) layer and reshape it as $X \in \mathbb{R}^{n \times d}$, where $n = H' \times W', d = C$. $X$ is transformed into $Q, K,$ and $V$, and relative positional biases $B(i, j)$ through linear projections. We define the attention weight for the $i$-th input within a region of size $k$ as the dot product between the query projection $Q$ of the $i$-th input and the key projections $K$ of the $k$ elements in the surrounding region:

$$A_i^k = \begin{bmatrix} Q_i K_{\rho_1(i)}^T + B_{(i, \rho_1(i))} \\ Q_i K_{\rho_2(i)}^T + B_{(i, \rho_2(i))} \\ \vdots \\ Q_i K_{\rho_k(i)}^T + B_{(i, \rho_k(i))} \end{bmatrix}, \tag{3}$$

where $\rho_j(i)$ denotes $i$'s $j$-th element in the region. We then define values, $V_i^k$, as a matrix whose rows are $k$ value projections from elements which in the region of $i$-th input :

$$V_i^k = \begin{bmatrix} V_{\rho_1(i)}^T & V_{\rho_2(i)}^T & \cdots & V_{\rho_k(i)}^T \end{bmatrix}^T. \tag{4}$$

Regional Attention for the $i$-th token with region size $k$ is then defined as:

$$NA_k(i) = \text{softmax}\left(\frac{A_i^k}{\sqrt{d}} V_i^k\right), \tag{5}$$

where $\sqrt{d}$ is the scaling parameter. This operation is repeated for every pixel in the feature map. Finally, the output from RAM is subjected to CA for global information interaction and feature fine-tuning. RAM can be represented as follows:

$$\tilde{X} = \text{CA}(\text{RAM}(\text{LN}(X))) + X, \tag{6}$$

$$\hat{X} = \text{GELU}(\text{MLP}(\text{LN}(\tilde{X}))) + \tilde{X}, \tag{7}$$

*3.2.3 Regional Attention With Larger Receptive Field.* Shadows are usually not isolated, they are intimately connected with their surroundings. A larger receptive field can enable a model to more comprehensively understand the relationships between shadowed and non-shadowed areas in an image, such as the shapes and edges of shadows and their relation to lighting conditions. If the receptive field is too small, the model might only see parts of the shadow or objects obscured by the shadow, potentially leading to incorrect shadow perception and removal. A larger receptive field allows the model to observe the entire shadow area and its transitional edges, thus more accurately performing shadow removal. During shadow removal, maintaining the naturalness and coherence of the image is crucial. A larger receptive field helps the model maintain consistency and natural transitions in the surrounding environment while removing shadows, and avoiding unnatural patches or color discrepancies in the processed image. Similarly, regional attention benefits from a larger receptive field, as tokens calculated within this area can access more information.

Inspired by DiNAT [14], to balance model complexity and performance, we propose a regional attention mechanism with a dilation factor. Specifically, we expand the receptive field to a greater range by increasing the stride when selecting regions, thereby maintaining the overall attention span. Using the regional attention mechanism with the dilation factor allows us to extend the receptive field of regional attention without increasing model complexity, further enhancing performance.

## 3.3 Loss Function

We employ two loss terms: content loss, and perceptual loss. We provide a detailed description of these loss terms below.

**Content Loss.** The content loss ensures consistency between the output image and the ground truth training data. In the image domain, we adopt the Charbonnier Loss [23]. The content loss can be expressed as:

$$\mathcal{L}_{cont} = \sqrt{(\hat{I} - I_{gt})^2 + \epsilon}, \tag{8}$$

where the $\hat{I}$ is the output image and $I_{gt}$ is the ground truth shadow-free image. The $\epsilon$ is set to $10^{-6}$ to ensure numerical stability.

**Perceptual Loss.** Perceptual loss has been widely used in various image restoration and generation tasks to preserve the high-level features and semantic information of an image while minimizing the differences between the restored image and the ground truth. We minimize the $l_1$ difference between the feature of $\hat{I}_{lit}$ and $I_{lit}$ in the {conv1_2, conv2_2, conv3_2, conv4_2, conv5_2} of a imagenet-pretrained VGG-19 model. Denoting the $i$-th feature extractor as $\Psi_i(\cdot)$, the perceptual loss we adapt can be expressed as follows:

$$\mathcal{L}_{per}^{D} = \sum_i w_i \|\Psi_i(I_{lit}) - \Psi_i(\hat{I}_{lit})\|_1, \tag{9}$$

where $w_i$ is the weight among different layers, the value of which is empirically set as {0.1, 0.1, 1, 1, 1}.
The total loss function turns out to be :

$$\mathcal{L} = \alpha_1 \mathcal{L}_{per} + \alpha_2 \mathcal{L}_{cont}, \tag{10}$$

where $\alpha_1, \alpha_2 = \{0.001, 1\}$ are empirically set. RASM undergoes end-to-end supervised training using the loss function $\mathcal{L}$, achieving state-of-the-art results in shadow removal.

## 4 EXPERIMENTS

### 4.1 Implementation Details

The proposed model is implemented using PyTorch. We train our model using AdamW optimizer [31] with the momentum as (0.9, 0.999) and the weight decay as 0.02. The initial learning rate is set to $4 \times 10^{-4}$, then gradually reduces to $10^{-6}$ with the cosine annealing [28]. We set the region size of the regional attention to 11 and the dilation factor to 2 in our experiments. Our RASM adopts an encoder-decoder structure ($L = 3$). We set the first feature embedding dimension as $C = 32$. During the training stage, we employed data augmentation techniques, including rotation, horizontal flipping, vertical flipping, MixUp [37], and adjustments in the H and S components of HSV color space.

To validate the performance of our model, we conduct experiments on two datasets. SRD [32] is a paired dataset with 3088 pairs of shadow and shadow-free images. We use the predicted masks that are provided by DHAN [2]. For the adjusted ISTD [24] dataset, we use 1330 paired shadow and shadow-free images for training and 540 for testing.

Following previous works [3, 13, 18, 34], we conduct the Root Mean Square Error (RMSE) between output image and ground-truth shadow image in the color space of CIE LAB as a quantitative metric (the lower the better). To make the comparison more comprehensive, we also follow [11] to report the Peak-Signal-to-Noise Ratio (PSNR) and Structural Similarity (SSIM) in RGB space (the higher the better). The FLOPs are reported on $256 \times 256$ images.

### 4.2 Performance Evaluation

*4.2.1 Quantitative Comparisons.* We first compare our proposed method with the state-of-the-art shadow removal methods on the ISTD+ [24] dataset. The competitors are DC-ShadowNet [19], BM-Net [40], DHAN [3], AutoExposure [9], G2R [3], ShadowFormer [11], ShadowDiffusion [12], Li et al. [25] and Liu et al. [27]. The input image is all resized to $256 \times 256$ for benchmarking following most of the existing methods [11, 13, 18, 41]. The results are depicted in Tab. 1. Our approach surpasses existing methods across all metrics in comparisons of Shadow Region, All Image, and also in the PSNR and RMSE metrics for Non-Shadow Regions, achieving SOTA performance. In Non-Shadow Regions, our SSIM is essentially on par with the best-reported results.

We also compare our method with the state-of-the-art shadow removal methods on the SRD [32] dataset. The competitors are consist of 11 methods, DSC [17], DHAN [3], AutoExposure [9], DC-ShadowNet [19], Unfolding [41], BMNet [40] ShadowFormer [11], ShadowDiffusion [12], Li et al. [25], Liu et al. [27] and DeS3 [20]. Since there exists no ground-truth mask to evaluate the performance in the shadow region and non-shadow region separately, we use a mask extracted from DHAN [3] following existing method [11] for comparison. The results are depicted in Tab. 2. As shown in Tab. 2, our method outperforms existing techniques across all metrics for the Shadow Region. Performance in the Non-shadow region does not quite match that of ShadowDiffusion [12] and Liu

| Method | Shadow Region (S) | | | Non-Shadow Region (NS) | | | All Image (ALL) | | |
|---|---|---|---|---|---|---|---|---|---|
| | PSNR↑ | SSIM↑ | RMSE↓ | PSNR↑ | SSIM↑ | RMSE↓ | PSNR↑ | SSIM↑ | RMSE↓ |
| DHAN [3] | 33.08 | 0.988 | 9.49 | 27.28 | 0.972 | 7.39 | 25.78 | 0.958 | 7.74 |
| G2R [3] | 33.88 | 0.978 | 8.71 | 35.94 | 0.977 | 2.81 | 30.85 | 0.946 | 3.78 |
| DC-ShadowNet [19] | 32.20 | 0.977 | 10.83 | 34.45 | 0.973 | 3.44 | 29.17 | 0.939 | 4.70 |
| AutoExposure [9] | 36.02 | 0.976 | 6.67 | 30.95 | 0.88 | 3.84 | 29.28 | 0.847 | 4.28 |
| BMNet [40] | 38.17 | 0.991 | 5.72 | 37.95 | **0.986** | 2.42 | 34.34 | 0.974 | 2.93 |
| ShadowFormer [11] | 39.67 | 0.992 | 5.21 | 38.82 | 0.983 | 2.30 | 35.46 | 0.973 | 2.80 |
| ShadowDiffusion [12] | 39.82 | - | 4.90 | 38.90 | - | 2.30 | 35.72 | - | 2.70 |
| Li et al. [25] | 38.46 | 0.989 | 5.93 | 37.27 | 0.977 | 2.90 | 34.14 | 0.960 | 3.39 |
| Liu et al. [27] | 38.04 | 0.990 | 5.69 | 39.15 | 0.984 | 2.31 | 34.96 | 0.968 | 2.87 |
| Ours | **40.73** | **0.993** | **4.41** | **39.23** | 0.985 | **2.17** | **36.16** | **0.976** | **2.53** |

Table 1: The quantitative results on ISTD+ [24] dataset. The best result is in bold, while the second-best one is underlined. To make a fair comparison, we use results published by the authors. − indicates that the metric is missed in the original paper.

| Method | Shadow Region (S) | | | Non-Shadow Region (NS) | | | All Image (ALL) | | |
|---|---|---|---|---|---|---|---|---|---|
| | PSNR↑ | SSIM↑ | RMSE↓ | PSNR↑ | SSIM↑ | RMSE↓ | PSNR↑ | SSIM↑ | RMSE↓ |
| DSC [17] | 30.65 | 0.960 | 8.62 | 31.94 | 0.965 | 4.41 | 27.76 | 0.903 | 5.71 |
| DHAN [3] | 32.71 | 0.943 | 6.60 | 33.88 | 0.949 | 3.46 | 29.72 | 0.923 | 4.40 |
| AutoExposure [9] | 31.34 | 0.933 | 7.90 | 29.74 | 0.916 | 5.21 | 26.99 | 0.869 | 5.95 |
| DC-ShadowNet [19] | 32.10 | 0.927 | 6.91 | 33.48 | 0.936 | 3.66 | 29.35 | 0.902 | 4.61 |
| BMNet [40] | 33.81 | 0.940 | 7.44 | 34.91 | 0.946 | 5.99 | 30.68 | 0.923 | 3.92 |
| Unfolding[41] | 34.94 | 0.980 | 7.44 | 35.85 | 0.982 | 3.74 | 31.72 | 0.952 | 4.79 |
| ShadowFormer [11] | 36.91 | 0.989 | 5.90 | 36.22 | 0.989 | 3.44 | 32.90 | 0.958 | 4.04 |
| ShadowDiffusion [12] | 38.72 | 0.987 | 4.18 | 37.78 | 0.985 | 3.44 | **34.73** | 0.970 | 3.63 |
| Li et al. [25] | 39.33 | 0.984 | 6.09 | 35.61 | 0.967 | 2.97 | 33.17 | 0.938 | 3.83 |
| Liu et al. [27] | 36.51 | 0.983 | 5.49 | 37.71 | 0.986 | 3.00 | 33.48 | 0.967 | 3.66 |
| DeS3 [20] | 38.73 | 0.987 | 4.70 | **38.12** | **0.988** | **2.72** | 34.19 | 0.968 | 3.59 |
| Ours | **40.26** | **0.993** | **3.90** | 36.80 | 0.987 | 3.19 | 34.46 | **0.976** | **3.37** |

Table 2: The quantitative results on SRD [32] dataset. The best result is in bold, while the second-best one is underlined.

| Method | Params (M) | GFLOPs | RMSE |
|---|---|---|---|
| DHAN [3] | 16.4 | 126.0 | 4.40 |
| AutoExposure [9] | 19.7 | 53.0 | 5.95 |
| BMNet [40] | **0.4** | **11.6** | 3.92 |
| Unfolding [41] | 10.1 | 48.2 | 4.79 |
| ShadowFormer [11] | 11.4 | 63.1 | 4.04 |
| ShadowDiffusion [12] | 55.2 | 896.7 | 3.63 |
| Li et al. [25] | 23.9 | 68.3 | 3.83 |
| Ours | 5.2 | 25.2 | **3.37** |

Table 3: Efficiency evaluation. Parameters count and GFLOPs are metered with fvcore[36] on 256×256 inputs. The best result is in bold, and the second-best result is underlined.

et al. [27], possibly due to stricter adherence to the shadow mask guidance. It is anticipated that our method would achieve a more satisfying result by employing a higher-quality shadow mask or utilizing user-provided masks. Notably, despite imprecise masks, our method is still the best among the competitors under RMSE for All Image.

To validate the efficiency of our method, we also conduct a comparison of FLOPs and parameter counts. As depicted in Tab. 3, our model has a small number of parameters and low FLOPs, utilizing a negligible amount of computational resources while achieving superior performance, demonstrating that our model effectively balances model complexity and model performance.

*4.2.2 Qualitative Comparisons.* This part exhibits several examples from SRD and ISTD+ datasets to compare the visual quality shown in Figs. 4 and 5. Our method achieves state-of-the-art performance with fewer residual shadow components and no visual artifact. Moreover, our total parameters are significantly fewer than most of the previous arts, which demonstrates the efficacy of our practical designs.

## 4.3 Model Analysis

**Discussion on Regional Attention and Window Attention.** To validate that our proposed regional attention is more suitable for the shadow removal task than traditional window attention, we choose the baseline model and its variants for comparison. Specifically, we replace all the window attention in the baseline model with regional attention, maintain the same area size of regional attention and window attention, and represent them as {Window Att., Regional Att.}. As shown in Tab. 4, the regional attention selected outperforms window-based attention on all three metrics, proving the superiority of our design.

**Discussion on Receptive Field of Regional Attention.** The size of the receptive field and the final effect of the shadow removal task

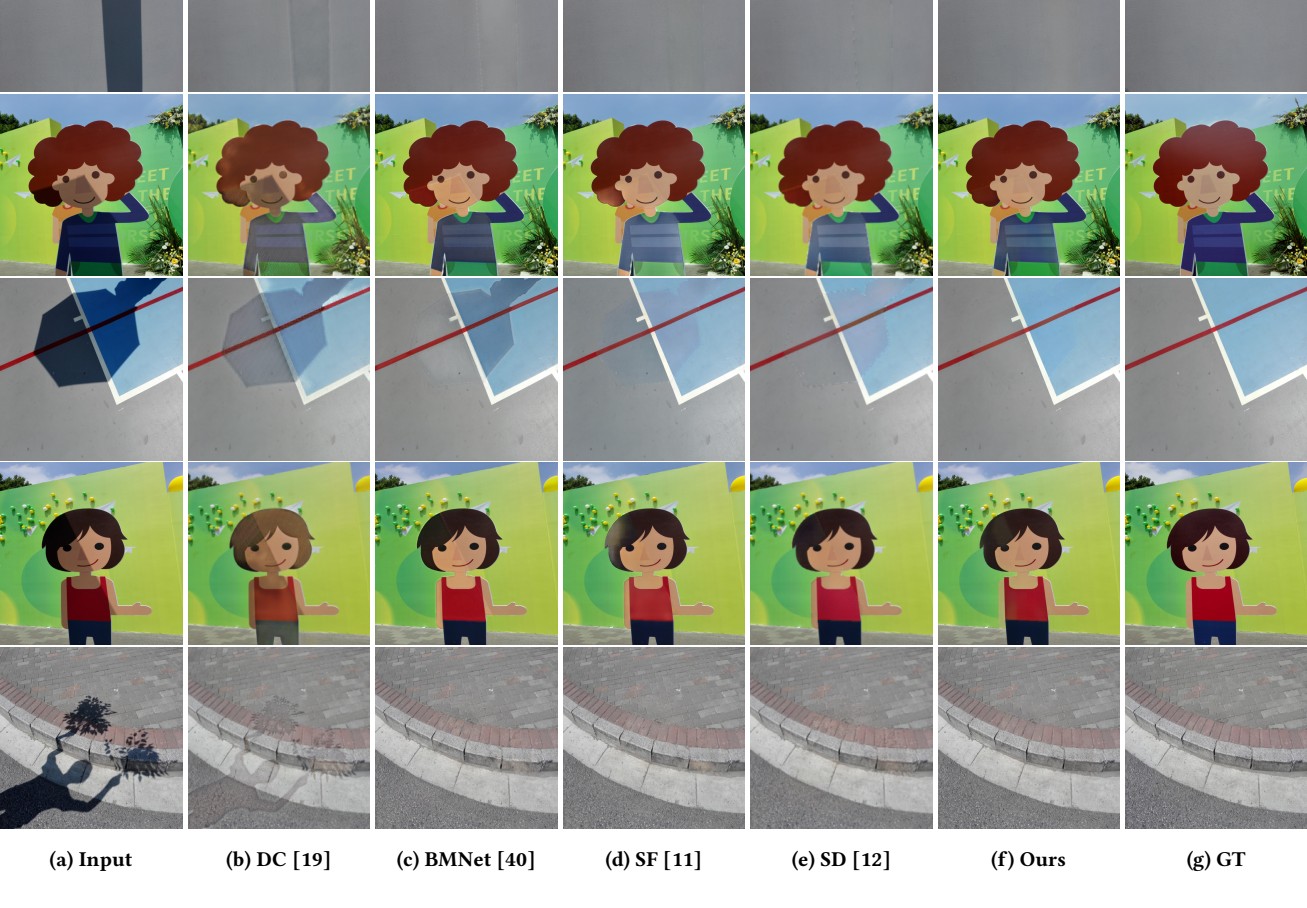

|         (a) Input |         (b) DC [19] |         (c) BMNet [40] |         (d) SF [11] |         (e) SD [12] |         (f) Ours |         (g) GT |

**Figure 4: Qualitative comparison on ISTD+ dataset. Please zoom in for more details. Due to page limits, we only exhibited 5 cases here, more cases can be found in the supplementary file.**

| Variant | Shadow Region (S) | | | All Image (ALL) | | |
|---|---|---|---|---|---|---|
| | PSNR↑ | SSIM↑ | RMSE↓ | PSNR↑ | SSIM↑ | RMSE↓ |
| *Window Att.* | 40.06 | 0.992 | 4.84 | 35.78 | 0.975 | 2.62 |
| *Regional Att.* | **40.73** | **0.993** | **4.41** | **36.16** | **0.976** | **2.53** |

**Table 4: Experiments for modeling on ISTD+ dataset. The best result is in bold.**

| Region Size | Dilation | PSNR | SSIM | RMSE | GFLOPs |
|---|---|---|---|---|---|
| $7 \times 7$ | 1 | 35.74 | 0.974 | 2.67 | 24.7 |
| $11 \times 11$ | 1 | 35.94 | 0.976 | 2.56 | 25.2 |
| $15 \times 15$ | 1 | 36.01 | 0.976 | 2.60 | 26.0 |
| $21 \times 21$ | 1 | 36.00 | 0.976 | 2.60 | 27.6 |
| $11 \times 11$ | 1 | 35.94 | 0.976 | 2.56 | 25.2 |
| **$11 \times 11$** | **2** | **36.16** | **0.976** | **2.53** | **25.2** |
| $11 \times 11$ | 3 | 36.02 | 0.976 | 2.59 | 25.2 |

**Table 5: Experiments for region size and dilation rate on ISTD+ dataset. Our final choice is marked in bold.**

are closely related. Here, we discuss two prominent parameters that control the receptive field size of our proposed regional attention mechanism: the size of the region $rs$ and the dilation factor $d$. We tried different region sizes and dilation factors to see how they affect the results.

As shown in Tab. 5, we found that as the $rs$ increases, the performance of the model improves while the computational load also increases. When $rs$ is greater than or equal to 15, the benefits obtained by adjusting $rs$ approach saturation. When $d$ is within the appropriate range, the model benefits most. However, when $d$ is too small, the spatial attention receptive field is limited. Conversely, when $d$ is too large, the spatial attention will choose a sparse distribution of region elements, making it difficult to aggregate non-shadow information from the surrounding shadowed area,

leading to performance degradation. To balance performance and computational complexity, we choose a regional attention size of $11 \times 11$ and a dilation rate of 2 for our model.

**Visualization of our regional attention.** To validate whether our proposed regional attention mechanism truly enables shadow areas to interact with their adjacent non-shadow areas, we selected several points on the image and visualized their attention weight

**Figure 5: Qualitative comparison on SRD dataset. Please zoom in for more details. Due to page limits, we only exhibited 5 cases here, more cases can be found in the supplementary file.**

| (a) Input | (b) DC [19] | (c) BMNet [40] | (d) SD [12] | (e) DeS3 [20] | (f) Ours | (g) GT |

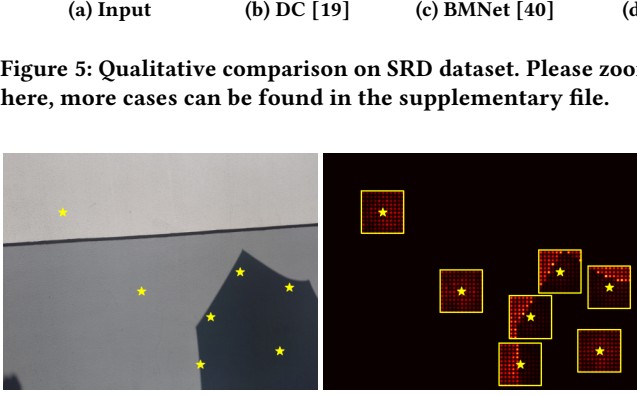

**Figure 6: A visualization of our regional attention. The original image is on the left, and the star marks the selected points. The heatmaps indicate the regional attention weight of the marked tokens. Brighter colors indicate a larger attention score.**

allocation. As shown in Fig. 6, we can see that in completely illuminated areas or shadow areas, the attention weights of these points are relatively low and even, while points that notice shadows have a much larger attention weight when the attention area can encompass the surrounding non-shadow areas. Moreover, points

with different shadow positions are paying attention to different non-shadow areas, corresponding to the fact that each shadow area information interacts with the adjacent non-shadow area information, which is consistent with our proposed region-based attention mechanism.

## 5 CONCLUDING REMARKS

In this work, we rethought the most significant information source for shadow removal, namely, the non-shadow areas adjacent to the shadow region, which plays a crucial role in this task. Based on this, we proposed a novel regional attention mechanism and introduced a lightweight regional attention-based shadow removal model, RASM. The regional attention mechanism introduced by us enables each shadow region to focus on specific information from surrounding non-shadow areas, thereby effectively utilizing this information for shadow removal. We demonstrated that RASM strikes a good balance between model complexity and model performance. Our model uses fewer parameters, lower FLOPs computation, and achieves superior performance on SRD and ISRD datasets.

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

Received 20 February 2007; revised 12 March 2009; accepted 5 June 2009

