# OpenReview forum: "Regional Attention For Shadow Removal"
_acmmm.org/ACMMM/2024/Conference — MM2024 Poster_

### Official Review · Reviewer_FPRy · 2024-05-22

**Rating:** 2
**Confidence:** 4

**Summary:**

Based on the large size and high computational complexity of existing models, this paper proposes a lightweight yet accurate shadow removal framework for image shadow removal.

**Strengths:**

To achieve accurate shadow removal, this paper introduces a regional attention mechanism, which enables model to focus on the critical areas surrounding the shadows. The experimental results seem to demonstrate the effectiveness of the proposed method.
The motivation of this paper is good.

**Limitations:**

The motivation of this paper is good, but the technological innovation of the method is limited. The network designed by the paper are relatively simple. The network structures of the proposed channel attention module and regional attention module are simple superposition of some operations that lack innovation.

The method introduces a lightweight model. However, from the results in Table 2, we find that the proposed model does not obtain an optimal value when evaluating the lightweight. Does this indicate that the proposed method does not achieve the so-called lightweight?
Moreover, why does does this approach achieve lightweight, and where in the module does it promote this? I am puzzled by this part. The article doesn't describe these details.

Paper argues that the existing window attention mechanism generates the attention within a window. But from the description, I believe that the proposed regional attention mechanism is also implemented using windows, just with a larger receptive field. I do not find the advantage of this module in terms of network structure.

The ultimate goal of shadow removal task is can process image with complex scenes. Paper only gives results for shadow image with simple scenes. That's not what we want to see. Can this method handle shadow image with complex scenes? I suggest more experiments and discussion about this.

**Suitability:**

3

---

### Official Review · Reviewer_mxR4 · 2024-05-23

**Rating:** 4
**Confidence:** 4

**Summary:**

This paper proposes a lightweight and accurate framework for shadow removal in images. The authors analyze the characteristics of the shadow removal task and introduce a regional attention mechanism to effectively capture the key information required for shadow reconstruction. They customize a Regional Attention Shadow Removal Model (RASM) that leverages non-shadow areas to assist in restoring shadow regions. Experimental results demonstrate that the proposed method outperforms other state-of-the-art models in terms of accuracy and efficiency.

**Strengths:**

- The proposed method outperforms existing shadow removal methods across various metrics on datasets like ISTD+ and SRD, achieving state-of-the-art performance.
- The model has a small number of parameters and low computational requirements, striking a balance between complexity and performance.
- The regional attention mechanism introduced in the model allows for effective utilization of information from non-shadowed areas for accurate shadow removal.

**Limitations:**

- Some typos should be clearly corrected.
- I suggest a visual example of the attention mechanism proposed in the article for visualization purposes.
- Is there any limitation to the methods in the article? Do the methods work for all scenarios?

**Suitability:**

2

---

### Official Review · Reviewer_PgW8 · 2024-05-25

**Rating:** 4
**Confidence:** 4

**Summary:**

The paper presents a lightweight and accurate framework for shadow removal. The proposed Regional Attention Shadow Removal Model (RASM) introduces a novel regional attention mechanism that leverages non-shadow areas to assist in restoring shadow regions. This approach ensures rational interactions and better contextual correlations between shadow and non-shadow areas. Extensive experiments demonstrate the superior accuracy and efficiency of RASM compared to state-of-the-art models.

**Strengths:**

The proposed method demonstrated superior balance between efficiency and performance.
The paper is well-structured and easy to read.

**Limitations:**

The concept of region attention in lines 438-450 needs clarification. Specifically, how is the key region determined? In conventional self-attention, the key region is the entire image, while in Swin-attention, it is a small patch. How is it defined in region attention? Furthermore, how does this approach reduce computational complexity compared to conventional self-attention? The introduction emphasizes the significance of non-shadow areas for effective shadow removal. How does the proposed region attention mechanism incorporate this principle into its design?

In line 439, 'k' appears to represent the size of the region, whereas in line 449, 'k' seems to denote the key. Please clarify this discrepancy.

From Table 5, the GFLOPs do not increase significantly with region size, which contrasts with the distinct differences in GFLOPs among the compared methods in Table 3. This suggests that the attention region size may not be the primary factor affecting computational complexity. The proposed method's computational efficiency might not be due to the attention design as suggested in lines 170-172. Analyzing the computational load of each network component could provide more insight.

**Suitability:**

3

---

### Official Review · Reviewer_YExQ · 2024-05-27

**Rating:** 4
**Confidence:** 4

**Summary:**

"Regional Attention for Shadow Removal" introduces a novel Regional Attention Shadow Removal Model (RASM).

The key innovation is the regional attention mechanism, which integrates information from non-shadow areas adjacent to shadows, balancing accuracy and computational efficiency.

**Strengths:**

1.  The regional attention mechanism is a novel, theoretically sound method that enhances shadow removal by focusing on locally relevant information.

2. The authors provide a clear rationale for the regional attention mechanism, supported by detailed problem analysis and ablation studies.

**Limitations:**

1.  The added complexity of the new attention mechanism is not thoroughly explored in terms of trade-offs between performance gains and complexity.

2. In Fig.5 in the supplementary, does it mean the method relies on mask accuracy?

3. There are missing citations. The following papers discuss the intrinsic property of shadow removal.
Estimating Reflectance Layer from A Single Image: Integrating Reflectance Guidance and Shadow/Specular Aware Learning

**Suitability:**

3

---

### Meta-Review · Area_Chair_Jd24 · 2024-07-04

**Recommendation:** Accept (Poster)
**Confidence:** 3

**Metareview:**

This paper introduces the Regional Attention Shadow Removal Model (RASM), a lightweight and accurate framework for shadow removal. RASM uses a novel regional attention mechanism to focus on critical areas around shadows, leveraging non-shadow areas to aid in restoring shadow regions. This approach ensures better contextual interactions between shadow and non-shadow areas. The experiments show that RASM is an accurate and efficient model. However, few concerns of the reviewers' are very crucial. First, the novelty of the method is limited, as pointed out by the reviewer. The presented method utilizes off-the-shelf components to remove shadow from images, which limits the technical advancement of the paper. Second, experiments to assess the model's performance in complex scenes are not presented. Third, the limitations of the presented method should be discussed along with the future research directions. The last is not a major concern but it is particularly mentioned by a reviewer.